# Individuals with latent tuberculosis in a high TB endemic country show mild COVID-19

Uzair Abbas[1,2], Kiran Iqbal Masood[1], Tulaib Iqbal[1], Bushra Jamil[3], Shama Qaiser[1], Maliha Yameen[1], Martin Rottenberg[4], Rabia Hussain[1], Zahra Hasan[1*]

**1** Department of Pathology and Laboratory Medicine, Aga Khan University, Karachi, Pakistan, **2** Department of Physiology, Dow University of Health Sciences, Karachi, Pakistan, **3** Department of Infectious Diseases, Aga Khan University, Karachi, Pakistan, **4** Department of Microbiology, Tumor and Cell biology, Karolinska Institute, Stockholm, Sweden

* zahra.hasan@aku.edu

## Abstract

### Introduction

Infection with *Mycobacterium tuberculosis* (MTB) may result in active tuberculosis (TB), bacterial clearance, or asymptomatic latent infection (LTBi). During the COVID-19 pandemic, interactions between MTB and SARS-CoV-2 infections were coincident in high TB burden countries such as Pakistan. The impact of LTBi on COVID-19 is not well understood. Here we investigated the association of LTBi with COVID-19 and its severity by determining cellular activation to MTB, IgG antibody responses and expression of genes associated with host immunity.

### Methods

Age and sex matched Healthy Controls (HC, n = 147) and COVID-19 patients (n = 128) were recruited in this cross-sectional study. COVID-19 was categorized as ambulatory (n = 103) or hospitalized (n = 25) disease. LTBi was determined using the X.DOT-TB ELISpot assay. RT-PCR based mRNA levels of IFN-γ, IFN-α, IL-6, IL-10, OAS1, MAVS, SOCS1 and SOCS3 were determined in PBMCs. IgG to SARS-CoV-2 and rubella virus were measured.

### Results

We found that 18% of COVID-19 patients and 32% of HC were LTBi positive (p < 0.0001). All COVID-19 LTBi positive cases had ambulatory disease. Logistic regression analysis revealed individuals with LTBi to have a 54% lower risk of COVID-19. The frequency of MTB-specific IFN-γ producing T cells was lower in COVID-19 patients than in HC LTBi positive individuals (p = 0.0095). The presence of a BCG scar was not associated with the occurrence of COVID-19.

**Data availability statement:** All relevant data are within the paper and its Supporting Information files.

**Funding:** TB Healthcare Biotechnology, China provided reagents and CDC, China provided support for the study. Uzair Abbas was supported by Higher Education Commission, Pakistan- Grand Challenges Fund Grant number -913, which also provided research support for this study.

**Competing interests:** The authors have declared that no competing interests exist.

Levels of IgG antibodies to SARS-CoV-2 were raised in COVID-19 cases but did not differ by LTBi status in HC or COVID-19 groups. IgG levels to rubella virus were similar regardless of LTBi status in control and patient groups. COVID-19 cases displayed higher expression of mRNA levels of MAVS, OAS-1, and SOCS3 ($p < 0.05$). Further, OAS-1 expression was raised in LTBi positive COVID-19 group as compared to the LTBi positive HC group ($p = 0.019$).

## Conclusions

We observed that T cell reactivity to MTB was associated with milder COVID-19. Reduced severity of COVID-19 and higher OAS-1 gene expression in COVID-19 LTBi positive individuals suggest a protective effect in these individuals. Further studies are required to investigate the combined impact of MTB and SARS-CoV-2 infections in the host.

## Introduction

The coronavirus disease (COVID-19) pandemic was a public health concern because of its rapid global spread, with 677 million cases and 6.9 million deaths reported worldwide up to 10 March 2023 [1]. Interestingly, Pakistan with its 220 million population reported approximately 31,000 COVID-19 related deaths- lower mortality than in many other countries [2]. A reason for lower COVID-19 mortality may be the younger-aged population, with individuals aged 20–40 years constituting a majority of the Pakistan population [3]. In addition, a state of pre-existing immunity to COVID-19 could arise due to prior exposure to human coronaviruses and commensal pathogens [4–6].

Tuberculosis (TB) resulted in 1.23 million deaths in 2024 worldwide [7]. About 5–10% of individuals infected with *Mycobacterium tuberculosis* (MTB) develop an active disease while other exposed individuals may clear the infection or develop persistent asymptomatic infection termed as latent TB infection (LTBi). The overall global population harboring viable MTB has been calculated to range between 4.9% (95%UI: 4.4–5.6) and 7.7% [8].

The overlap between TB and COVID-19 is unavoidable in a disease endemic country such as Pakistan, which ranks 5th amongst high TB burden countries with an incidence of 331/100,000 cases annually [9]. The burden of TB in Pakistan can result in high community-based exposure with the frequency of LTBi shown to range from 30% to 70% in household contacts [10]. LTBi diagnostic assays such as the TST (tuberculin skin test) and IGRA (IFN-γ release assays) measure T cell responses to MTB antigens but do not discriminate between past or current infection [11].

In high infectious disease burden settings, the co-occurrence of different infections may add to the complexity of outcomes due a particular pathogen. MTB is shown to downregulate both innate and adaptive arms of the immune system. TB is associated with a decreased T cell response and dysregulation of pro-and anti-inflammatory cytokine balance in the host [12]. A state of enhanced inflammation with upregulated

inflammatory cytokines and interferon-pathway regulatory Suppressor of Cytokine Signaling (SOCS) molecules-1 and −3 are observed in LTBi [13]. Separately, restriction of SARS-CoV-2 infections requires activation of innate and adaptive immunity. Asymptomatic as compared with severe COVID-19 infections are characterized by more effective anti-viral immunity mediated by interferon responses [14]. The activation of interferon-stimulated genes (ISG) such as, 2'-5'-oligoadenylate synthetase (OAS1) and mitochondrial antiviral-signaling protein (MAVS) are associated with mild COVID-19 [15]. Whilst, severe COVID-19 is associated with excessive inflammation and a cytokine storm which may lead to acute respiratory distress and unfavorable outcomes [16].

We investigated whether immune modulation due to latent MTB infection may alter the consequence of SARS-CoV-2 infection. Currently, studies on the association between SARS-CoV-2 and MTB infections are both scarce and discordant [17,18]. Clinical evidence suggests that SARS-CoV-2 infection predisposes patients to TB infection, may lead to reactivation of latent disease or worsen COVID-19 severity [19–21].

Here, we studied MTB-induced T cell reactivity in a cohort of healthy controls (HC) and COVID-19 patients from Karachi, Pakistan. We determined the association between LTBi positivity with severity of COVID-19 as well as mRNA expression of immune related genes including those required for MTB as well as SARS-CoV-2 control; IFN-γ, IFN-α, IL-6, IL-10, OAS1, MAVS, SOCS1 and SOCS3 molecules. We also investigated humoral immunity through the measurement of antibody (IgG) response to SARS-CoV-2 and rubella virus in study participants. The study was carried out between 2022 and 2023, a time when the omicron variants were in circulation [22]. Our results suggest that the LTBi state is associated with less severe COVID-19.

## Materials and methods

### Ethical consideration

The study was approved by Ethical Review Committee (ERC) of the Aga Khan University (AKU) Karachi, Pakistan, with reference numbers 2021-6673-20227 and 2022-6871-22433. All participants were recruited with written informed consent. Hospitalized patients were consented directly or through their next of kin.

### Study description and selection of subjects

This was a cross-sectional study. Participants were recruited through consecutive convenience sampling from Aga Khan University Hospital (AKUH) between February 2022 and July 2023. We included both females and males aged 18 years and over. We excluded participants with a known history (based on verbal recollection) of acute viral infection or chronic (Hepatitis B/ C/ HIV) infection, active TB and/or pregnancy.

All study participants were tested using the Standard Q COVID-19 IgG/IgM combo assay (SD Biosensor, South Korea) at the time of enrolment in the study.

The HC group comprised individuals without any history of COVID-19 and comprised healthcare workers and community-based volunteers. All HC group participants tested negative using the COVID-19 test.

All participants in the COVID-19 group were confirmed by testing of respiratory samples using the reverse-transcription (RT) polymerase chain reaction (PCR)- test (Cobas SARS-CoV-2 assay; Roche diagnostics, USA).

COVID-19 patients who had a positive PCR result from the AKUH Clinical Laboratories, were contacted by the study team and consented to the study. Blood samples were collected through AKUH Clinical Laboratory or Home Health team. We recruited those with active as well as convalescent COVID-19. Active COVID-19 (COVID-19-a) cases were defined as those recruited within 72 h of a positive RT-PCR result. COVID-19 recovered (COVID-19-r) cases comprised individuals with a prior positive SARS-CoV-2 PCR test result, at least 12 weeks prior to recruitment with a median recovery period of 29 (IQR = 13) weeks. The COVID-19-a group tested Positive whilst the COVID-19-r group tested Negative on the COVID-19 IgG/IgM combo assay.

### Data collection

Data on age, gender, presence or absence of BCG (Bacillus Calmette-Guérin) scar, any history of comorbid conditions, known contact with TB patients, and COVID-19 vaccination status of individuals was recorded for each study participant.

### Sample collection

Blood was collected in heparinized tubes for the X.DOT-TB (TB Healthcare Biotechnology, China) and QuantiFERON TB Gold in Tube (QFT GIT, Qiagen, Cellestis, USA) assays. Blood was also collected in a gel-top tube for serum isolation and antibody testing.

### X.DOT-TB assay testing

Peripheral blood mononuclear cells (PBMCs) were separated by Ficoll-Histopaque method. The X.DOT-TB assay was performed as described previously [23]. Briefly, PBMCs were plated at $2.5 \times 10^5$/ well and cultured for 20 hours in plates coated with mouse anti-human IFN-γ. Each sample was incubated with Test (T, MTB antigens from ESAT-6 and CFP-10), a Positive control (P, mitogen control) and with an unstimulated Nil (N) well. Test wells were washed and incubated with mouse anti-human IFN-γ monoclonal HRP-labeled antibody for staining. Images were captured with a digital microscope and spots counted as Interferon-γ Spot Forming Units (IFN-γ-SPUs or the T score). Positive, negative, or indeterminate results were labeled with respect to the number of IFN-γ-SPUs in Nil and Test-wells. Sample results from the X.DOT-TB assay are shown in Fig 1A.

### Comparison of X.DOT-TB and QuantiFERON-TB Gold In-Tube assays

We compared the performance of X.DOT-TB assay with the QuantiFERON-TB Gold In-Tube (QFT-GIT) assay. QFT-GIT was used as per the manufacturer's instructions (Qiagen, Cellestis, USA). QFT-stimulated plasma was harvested after 20 hours and tested for IFN-γ concentrations (IU/ml). Positive samples were those with IFN-γ ≥ 0.35 IU/ml. Samples were labeled as indeterminate if the mitogen minus negative control value was < 0.5 IU/ml. Samples were found to be indeterminate in QFT testing and were excluded from the comparisons. Whole blood from 17 LTBi positive and 19 LTBi negative cases identified earlier by the X.DOT-TB assay were re-tested using the QFT-GIT assay with results shown in Fig 1B. Based on comparison with QFT results, the X.DOT-TB assay was found to have a sensitivity of 94.4%, specificity of 95%, a positive predictive value (PPV) of 94.4% and a negative predictive value (NPV) of 95% as (Fig 1C).

### mRNA expression by RT-PCR

One million PBMCs were lysed using 500 µl Trizol reagent (Invitrogen, USA) and processed for RNA extraction. cDNA was synthesized from 200 ng RNA using the RevertAid First Strand cDNA Synthesis Kit (Thermo Fisher Scientific, USA). Each sample was tested in duplicate by using 2 µL cDNA and primers at 150nM each in the reverse-transcription (RT) PCR conducted with the SYBR® Green PCR master mix (Applied Biosystems, USA) using the Real-time PCR CFX Opus-96 (Bio-Rad, USA) equipment. Sequences of the primer used are provided in S1 Table while, IFN-γ, IL-6 and IL-10 primers were obtained from RT-PrimerDB (www.rtprimerdb.org). SOCS1 and SOCS3 primers were designed using primer express software (version 3.0, Applied Biosystems, USA). These were used as reported previously using HuPO as a control gene to calculate the relative amount of transcript using the $2^{-(\Delta\Delta Ct)}$ method to determine relative gene expression [24–26]. IFN-α, OAS1 and MAVS primer sequences were used as reported [27–29].

### Measurement of IgG to RBD protein

IgG to RBD was measured in the sera of study participants. Recombinant RBD protein was provided by the laboratory of Prof. Paula M. Alves, (Institute of Experimental Biology and Technology; iBET, Portugal). The soluble RBD protein includes

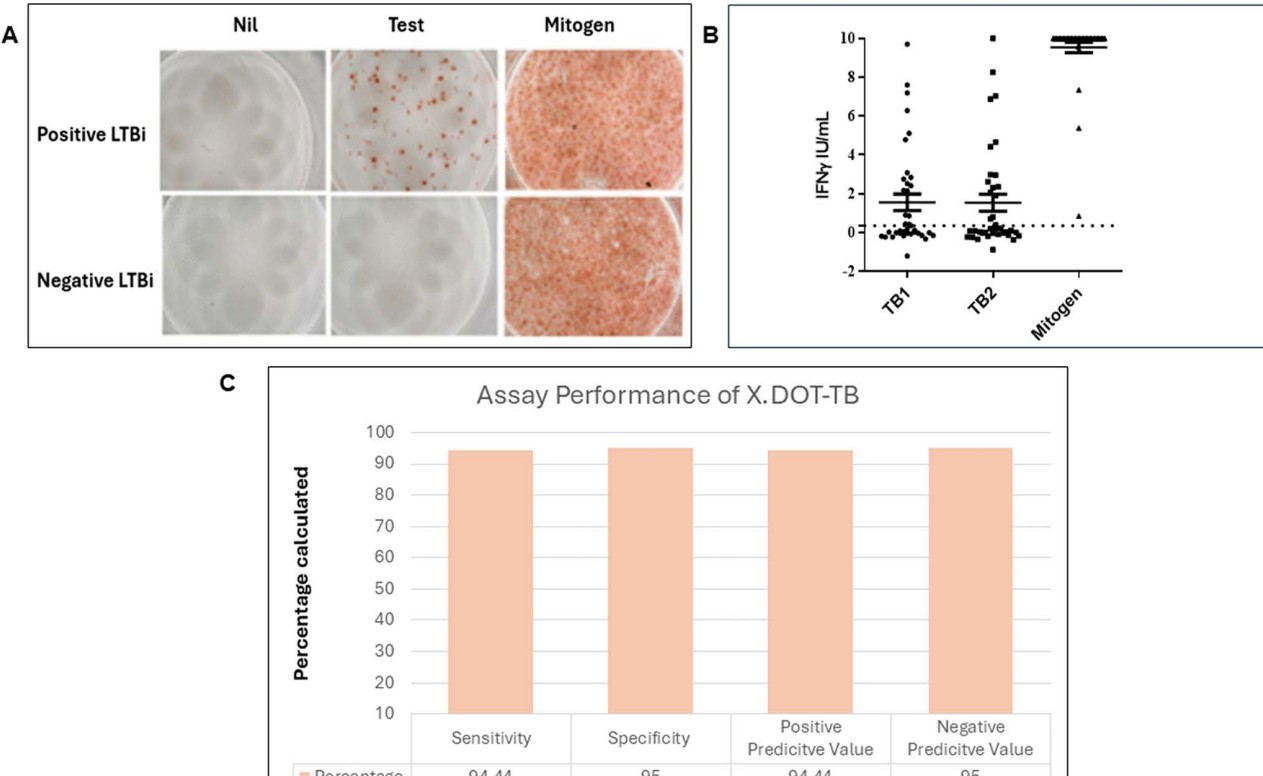

**Fig 1. Comparison of LTBi detection by X.DOT-TB assay and QFT-GIT assays.** A. The figure depicts the results of LTBi positive and negative samples detected by the X.DOT-TB assay which identified IFN-γ –positive cells in wells with Negative (NC), Test sample and Positive (mitogen control) wells. Spot forming units (SPUs) were counted in each case. Results were interpreted as per manufacturer's instructions with the cut-off for MTB positivity at Test-NC=≥10 SPU/well. B; QuantiFERON TB gold plus assay (QFT GIT assay). Values are shown as values of TB1, TB2 and MITOGEN (-Nil). The dotted line indicates the cut-off of positivity at 0.35 IU/ml. C; X.DOT-TB assay performance calculation using MedCalc for sensitivity, specificity, positive predictive value (PPV) and negative predictive value (NPV) against the QFT-GIT assay.

the signal peptide and C-terminal hexahistidine tag and has reported to be stable and consistent for use in serological assays [30] and was used for the detection of serum IgG by enzyme-linked immunosorbent assay (ELISA) as described earlier [31]. Briefly, SARS-CoV-2 RBD protein was coated at a concentration of 2 µg/ml, incubated with test sera, stained with goat anti-human IgG Fc (HRP), and developed for reading at $OD_{450}$ nm. The cut-off for positive responses of IgG to Spike and RBD was ≥ 0.5 $OD_{450}$ nm as established by validation against pre-pandemic negative control samples.

### Titration of IgG antibodies

Antibody titers to RBD of SARS-CoV-2 were determined by running ELISA assays under the conditions described above. For titrations, each serum sample was tested in six sequential 2-fold dilutions starting at 1:100. Results of the IgG $OD_{450}$ nm values obtained across the titrations for sera were used to plot titration curves for samples tested.

### Measurement of IgG to Rubella virus

IgG antibodies to rubella virus were measured in sera of study subjects. Antibody testing was performed at AKUH Clinical Laboratories using Electrochemiluminescence Immunoassay (ECLIA) on Cobas® e601, Roche systems. The cut-off for positive responses was 10 IU/ml.

## Data analysis

Statistical analysis was carried out using Statistical Packages for Social Sciences (SPSS) version 17 and GraphPad PRISM version 8 (GraphPad Software Inc. USA). The data was presented as median values. The Chi-squared test was used to compare categorical or qualitative variables between X.DOT-TB positive and negative groups. Logistic regression analysis was applied to study the adjusted odds ratio for association of LTBi with COVID-19 disease severity. The sensitivity, specificity, negative and positive predictive values of X.DOT-TB as compared with QuantiFERON TB Gold Plus was calculated using MedCalc (MedCalc Software Ltd, Belgium). The quantitative data were compared through the Mann-Whitney U-test. Antibody titers in study groups were compared using an Area Under the Curve (AUC) analysis.

## Results

### Characteristics of study participants

We first determined the rates of LTBi positivity as assessed by T cell -IFN-γ reactivity to MTB antigens in a cohort of healthy controls (HC) and participants who had suffered COVID-19. Two hundred and seventy-five study subjects were recruited between January 2022 and July 2023, with 147 study subjects in the HC and 128 individuals in the COVID-19 group. In the HC group, the mean age of individuals was 41.4 years and there were 51.7% females (Table 1). Amongst controls, 77% of individuals had a BCG scar and 15.6% had a history of contact with TB patients. We found no differences in gender, age, BCG scar or TB exposure history between HC and COVID-19 groups. However, the prevalence of comorbidities such as diabetes, hypertension, thyroid disorders, cancers, cardiac diseases, and asthma was higher in COVID-19 (33.6%) as compared with HC (12.9%) participants, p = 0.001. At the time of recruitment, 78.9% of individuals in the HC and 99.2% in the COVID-19 group had received COVID-19 vaccinations (p = 0.034).

### Comparison of frequency of LTBi between healthy controls and COVID-19 cases

A significantly higher frequency of LTBi positive individuals was observed in the HC (47/147; 32%) as compared to COVID-19 (23/128; 18%) group, (Table 1, p < 0.001 Chi $x^2$ test). There were some indeterminate results for LTBi: 3.4% (5/147) in the HC and 3.12% (4/128) individuals in COVID-19 group. The LTBi-positive individuals in both HC and COVID-19 groups showed similar frequencies of BCG scar, history of TB contact, and comorbid conditions (S2 Table). Moreover, logistic regression analysis revealed that individuals with LTBi had 54% less chance of having COVID-19 when age, gender, BCG vaccine and comorbid conditions were adjusted (Exp B = 0.46, p = 0.008).

**Table 1. Characteristics of study subjects.**

| Variables | | Healthy controls (n = 147) | COVID-19 cases (n = 128) | p value |
|---|---|---|---|---|
| Age (years) | Mean (SD) | 41.4 ± 11.57 | 42.4 ± 14.33 | 0.25# |
| Gender | Female | 76 (51.7%) | 67 (52.4%) | 0.995 |
| BCG scar | Yes | 108 (73.4%) | 94 (73.4%) | 0.547 |
| History of contact with TB cases | Yes | 23 (15.6%) | 17 (13.2%) | 0.579 |
| COVID-19 vaccination | Yes | 116 (78.9%) | 127 (99.2%) | 0.034* |
| **Comorbidities | Diabetes | 7 (4.7%) | 17 (13.3%) | 0.001* |
| | Others | 12 (8.2%) | 26 (20.3%) | |
| LTBi | Positive | 47 (32%) | 23 (18%) | <0.001* |

BCG, bacille Calmette-Guerin vaccination. **Other comorbidities included hypertension, thyroid disorders, cancers, cardiac diseases and asthma. Statistical analysis between groups was performed using either #Mann-Witney test, or the chi square test. *Differences are significant at p < 0.05 at 95% CI.

## Association of LTBi with severity of COVID-19

Next, we compared the frequency of LTBi in mild and severe COVID-19 individuals. COVID-19 cases were categorized as ambulatory or hospitalized patients as per the World Health Organization (WHO) ordinal scale of COVID-19 disease severity [32]. Cases included 99 participants with ambulatory and 25 with hospitalized COVID-19.

All twenty-three LTBi positive cases in the COVID-19 group had mild or asymptomatic disease and therefore LTBi showed a positive association with mild COVID-19 (p=0.008, Table 2). The comparison of age, gender, presence of a BCG scar, and contact with TB patients between ambulatory and hospitalized COVID-19 is shown in Table 2. The frequency of comorbidities within hospitalized COVID-19 individuals was higher than those with ambulatory disease (p<0.001 $x^2$ test).

## Comparison of *M. tuberculosis*-specific T cells between LTBi positive HC and COVID-19

The number of MTB-specific IFN-γ secreting T cells as measured by the X.DOT-TB assay amongst LTBi positive in COVID-19 and HC groups was compared. LTBi individuals within the HC displayed higher MTB-specific T cell frequencies than those with COVID-19 (Fig 2A, p=0.0095). We further compared MTB-specific IFN-γ secreting T cells between participants with active (COVID-19-a; n=45) or recovered (COVID-19-r; n=83) COVID-19. The rate of MTB- specific T cells between active and recovered COVID-19 cases was similar (Fig 2B). Notably, the numbers of MTB-specific IFN-γ secreting T cells were significantly higher in ambulatory as compared to hospitalized COVID-19 group (Fig 2C; p=0.022).

## Comparison of mRNA levels of immune-related genes between HC and COVID-19 groups

We determined mRNA expression levels of immune related genes in PBMCs in a sub-set of individuals from our study groups. Individuals in the COVID-19 group showed higher expression of OAS1 (p=0.0002), MAVS (p=0.0001) and SOCS3 transcripts (p=0.001) as compared to HC (Fig 3C-3D,3H). The increased expression of OAS1, MAVS and SOCS3 mRNA in COVID-19 was consistent for both active and recovered cases (S1 Fig).

**Table 2. Association of COVID-19 disease severity with X.DOT-TB cases.**

| Variables | | COVID-19 cases*** | Ambulatory | Hospitalized | p-value |
|---|---|---|---|---|---|
| | | n=124 | n=99 | n=25 | |
| LTBi status# | Positive | 23 | 23 (23.7%) | 0 (0.0%) | *0.008** |
| | Negative | 101 | 76 (76.3%) | 25 (100.0%) | |
| LTBi status as per COVID-19 recency* | Active | 43 | 5/24 (19.2%) | 0/19 | 0.232 |
| | Recovered | 81 | 18/75 (25.35%) | 0/6 | |
| Age | Median (min-max) | | 35 (29-44) | 59 (43-69) | *<0.001** |
| Gender | Female | 65 | 56 (56.7%) | 9 (39.1%) | 0.166 |
| BCG scar | Yes | 92 | 71 (71.1%) | 21 (82.6%) | 0.210 |
| History of TB contact | Yes | 17 | 13 (13.4%) | 4 (13.0%) | 0.709 |
| Comorbidities | None | 83 | 75 (75.3%) | 8 (26.1%) | *<0.001** |
| | Diabetes | 16 | 6 (7.2%) | 10 (43.5%) | |
| | Others | 25 | 18 (17.5%) | 7 (30.4%) | |

#LTBi determined by X.DOT-TB assay; *, recency of COVID-19 infection is shown, active COVID-19' as those with a positive SARS-CoV-2 test within 72 h of sample collection; Pearson's chi-squared test was used to assess the association of severity of disease with age, gender, BCG scar, history of TB contact, comorbid conditions and X.DOT-TB tests status. Differences are significant at **p ≤ 0.05 chi squared test. *** (n=4) individuals with indeterminate results of X.DOT-TB assay were excluded from analysis in this table.

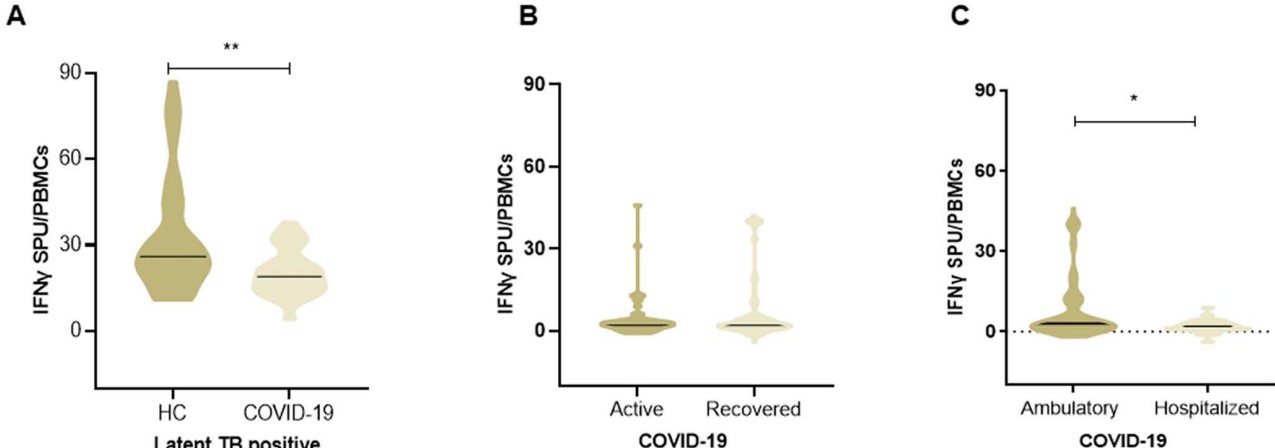

**Fig 2. Lowered MTB-specific IFN-γ responses in COVID-19 and those with severe disease.** The violin plots depict the spot forming units (SPU) show T cell -IFN-γ to MTB antigens through the X.DOT-TB assay. A, Comparison of LTBi positive cases in HC (n = 47) and COVID-19 (n = 23) previously identified using the X DOT TB assay. B, MTB- SPUs in COVID-19 active (n = 45) and recovered (n = 83) cases. C, Ambulatory (n = 103) and hospitalized (n = 25) sub-groups of COVID-19. The horizontal line depicts median values. The Mann-Whitney U test was used to calculate the statistically significant differences (p ≤ 0.05) between groups.

### Gene expression of positive LTBi in healthy controls and COVID-19 cases

To investigate any effect due to LTBi, we compared gene expression between HC and COVID-19 participants. A higher expression of OAS1 (p = 0.019) was observed in LTBi positive COVID-19 as compared with LTBi positive HC groups. Whilst mRNA levels of IFN- γ, IFN-1α, MAVS, IL-6, SOCS1 and SOCS3 were similar between LTBi positive individuals in the HC and COVID-19 groups (S2 Fig). Of note, mRNA levels of the target genes were comparable between LTBi positive and negative individuals in the HC and COVID-19 groups respectively (S3 Table).

### Comparison of COVID-specific or heterologous IgG antibody levels in study groups

We next investigated whether humoral immunity against SARS-CoV-2 was affected by LTBi status, we compared IgG antibodies to Spike (RBD) between LTBi positive and negative individuals. The IgG antibodies levels to RBD in sera of COVID-19 (n = 120) cases were higher than in the HC group (n = 60; p < 0.001, Fig 4A). Notably, antibody levels were similar between LTBi positive and negative individuals of both HC and COVID-19 groups (Fig 4B-4C). In addition to measuring serum IgG OD levels, we ran antibody titrations for a sub-set of sera from both COVID-19 and HC groups (n = 30 each respectively, Fig 4D-4E). AUC analysis of serum IgG titers to RBD confirmed higher levels COVID-19 as compared with the HC group (Fig 4F).

We also investigated whether the IgG levels to Rubella virus-a non-related viral antigen were affected by MTB in the study groups. The study population is vaccinated against Rubella virus infection and therefore it was not surprising to find 94% of individuals to be seropositive to Rubella virus. Anti-Rubella IgG levels between HC and COVID-19 groups were not significantly different (Fig 5A). Moreover, there was no difference in anti-Rubella IgG titers between LTBi positive or negative individuals within HC (Fig 5B) or COVID-19 (Fig 5C) groups.

### Discussion

Pakistan is a high TB burden country which experienced relatively low COVID-19 morbidity during the pandemic. We show a positive association between LTBi with mild COVID-19 disease. We found 18% LTBi positivity in COVID-19 as

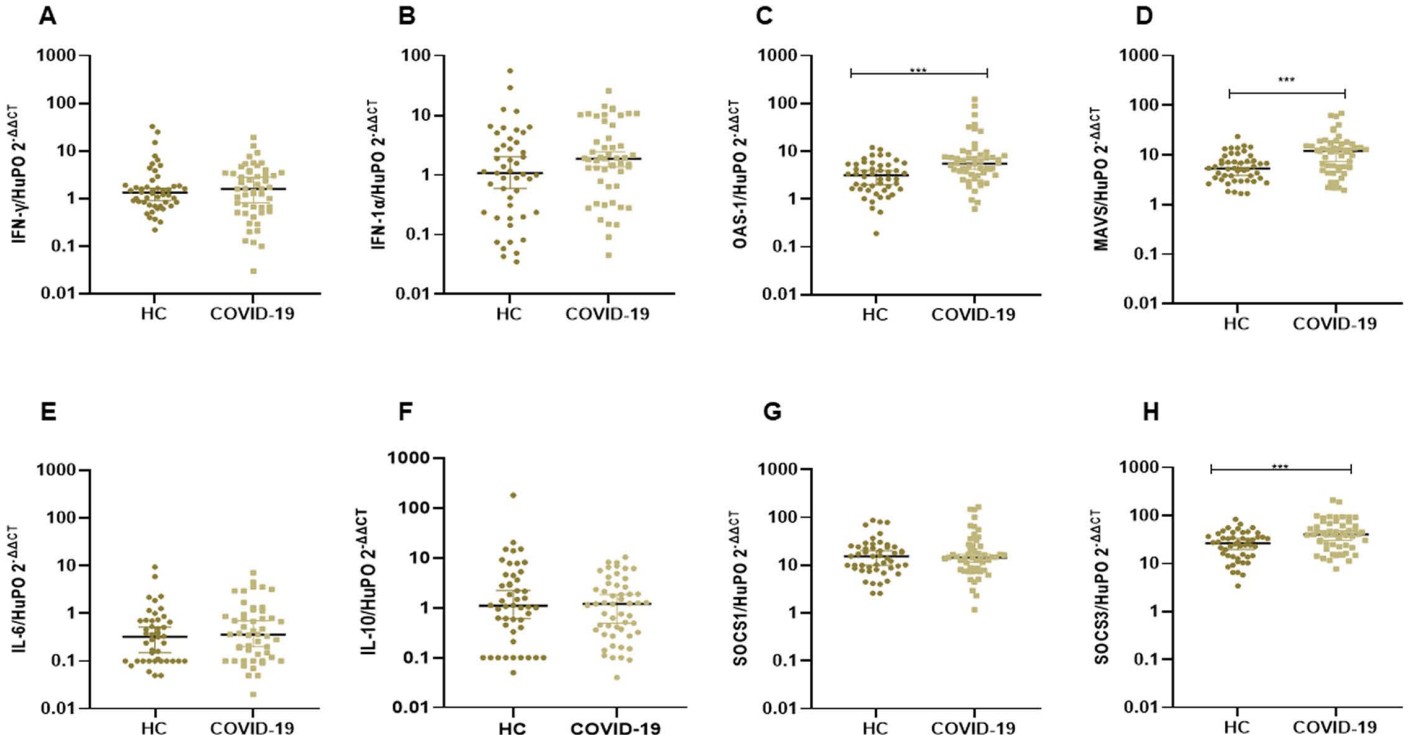

**Fig 3. Increased mRNA expression of OAS1, MAVS and SOCS3 in COVID-19 cases.** The figure shows comparison of mRNA expressions in unstimulated PBMCs from Healthy controls (n = 48) and COVID-19 cases (n = 52). The graphs show mRNA expression levels of, IFN-γ (A), IFN-α (B), OAS-1 (C), MAV (D), IL6 (E); IL10 (F); SOCS1 (G); and SOCS3 (H), Gene expression was normalized to HuPO and shown as median value of target gene/HuPO for each sample by the relative quantification method 2-ΔΔCT. '***' shows p < 0.0001, Mann-Whitney U test, comparing the fold change in gene expression.

compared to 32% in the control group. Furthermore, regression analysis showed individuals with LTBi had a 54% lesser risk of COVID-19.

Our data is concordant with earlier reports of LTBi among COVID-19 patients. A study from Florence, Italy, reported 10.9% of individuals with LTBi among 202 COVID-19 patients, and that there was less severe disease in MTB infected individuals [33]. Higher rates of LTBi were reported in India, whereby Madan et al. reported 25% of patients had LTBi among COVID-19 patients and those were found to have less severe COVID-19 [17].

The lower rates of LTBi identified in COVID-19 subjects may primarily arise from an increased heterologous immunity to SARS-CoV-2. In our study, identification of LTBi only in those with mild/asymptomatic COVID-19 disease suggests that Latent TB infection may protect against the development of severe infection. We observed that individuals who suffered severe COVID-19 showed lower levels of MTB-specific IFN-γ secreting T cells as compared to those who had mild or asymptomatic COVID-19. In individuals with active or recovered COVID-19, a diminished IFN-γ response MTB antigens may reflect the immunological imprint of severe viral diseases and the presence of comorbidities. Our data support previous reports that show SARS-CoV-2 and MTB infections can have a concomitant effect on the host [34].

Lower levels of MTB-specific T cells in COVID-19 group might be attributed to a downregulated immune response after acute COVID-19 [35]. COVID-19 can have a long-lasting impact on human health and growing evidence indicates that SARS-CoV-2 infection induces persistent alterations in immune function characterized by altered cytokine production and immune cell exhaustion, which may persist for months after viral clearance [36]. Reduced immune responses and

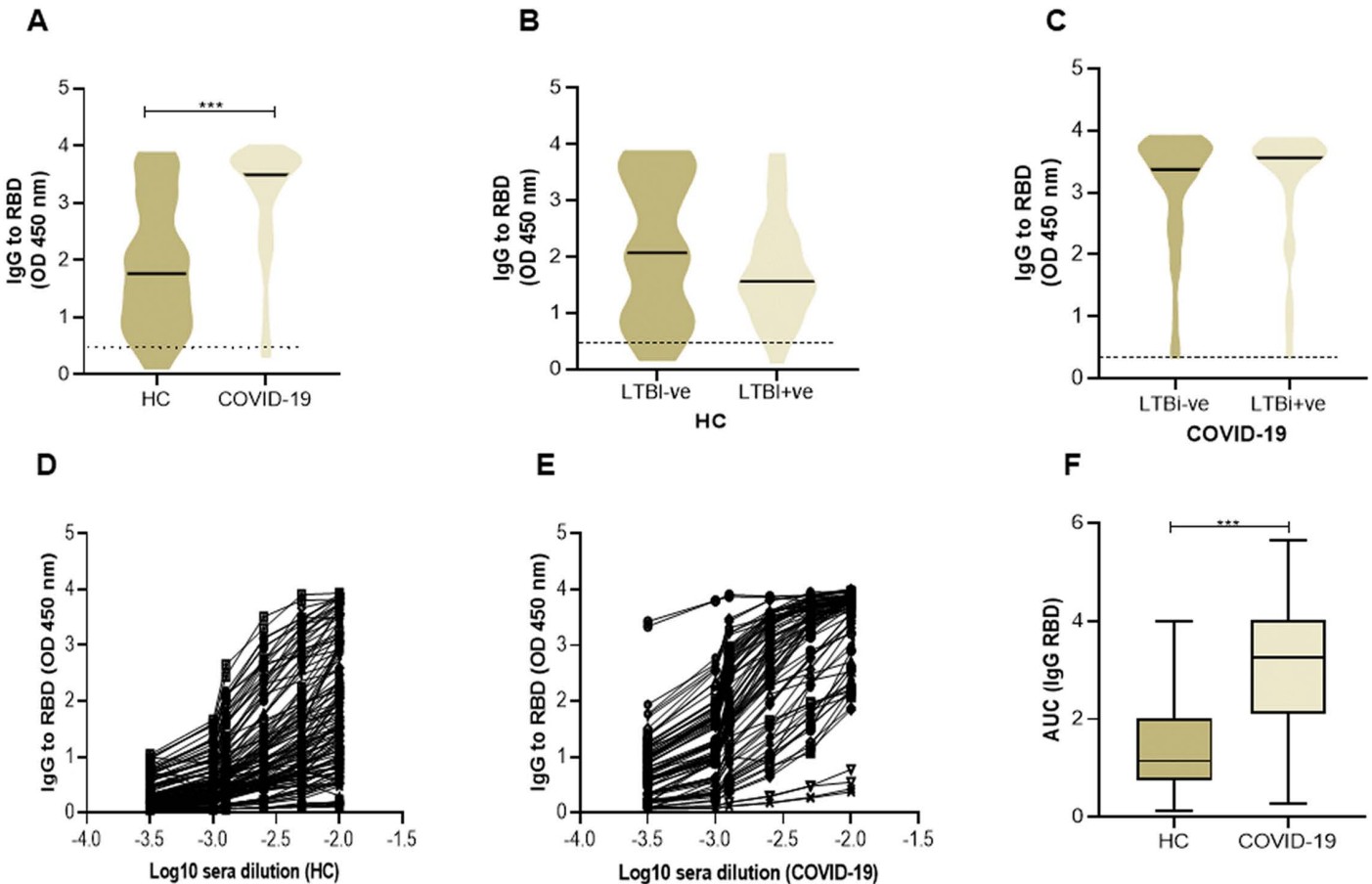

**Fig 4. Comparison of IgG to SARS-CoV-2 RBD in HC and COVID-19 groups.** We measured the serum levels of IgG to RBD protein in HC (n = 60) and COVID-19 (n = 120) groups. A) The violin plot depicts IgG levels in both groups. B) The graph compares IgG levels in HC who were either LTBi- (n = 30) or LTBi+ (n = 30). C) IgG levels in COVID-19 cases LTBi-ve (n = 97) or LTBi + ve (n = 23) are shown. The plot depicts the OD level with cut-off for positive responses OD ≤ 0.05) indicated by a dotted horizontal line. Titration curves of Log 10 dilution of serum samples for measurement of IgG levels against RBD protein of SARS-CoV-2 among healthy controls and COVID-19 cases is shown in D, Healthy controls (n=60), E, COVID-19 (n=75). For titrations, each serum sample was tested in six sequential 2-fold dilutions starting at 1:100 respectively.F) Comparison of AUC for IgG among healthy controls and COVID-19. The Mann-Whitney U test was used to calculate the statistically significant differences (p≤0.05) between groups.

diminished T cell frequencies after COVID-19 might have caused the decreased IFN- γ response against MTB antigens as observed earlier in COVID-19 cases [37].

HC and COVID-19 were comparable with regards to age and gender. We found similar gender and age distribution among LTBi in HC and COVID-19 groups. Most of the individuals recruited for this study showed a BCG scar, which was not associated with the frequency of COVID-19, suggesting that most likely, MTB exposure rather than BCG vaccination conferred relative protection observed against infection and development of mild COVID-19. The expanded program of immunizations (EPI) program in Pakistan delivers vaccinations against childhood diseases at birth, including, BCG vaccinations. Analyses from the early pandemic period described that there is no protective effect of BCG on COVID-19 risks [38,39]. It was however noted that COVID-19 related mortality was lower in high TB burden countries such as Pakistan, which had high BCG coverage [40]. Moreover, comorbidities were significantly higher in the COVID-19 patients and may have contributed to COVID-19 development as previously shown [41]. However, we did not find any association between

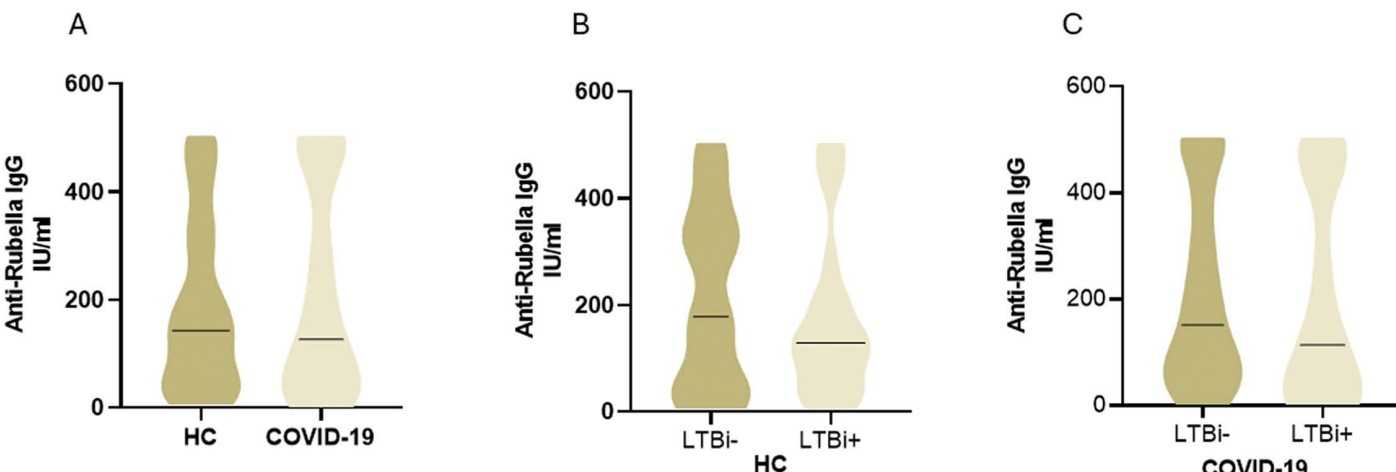

**Fig 5. Anti -Rubella virus IgG levels in study subjects in HC and COVID-19 groups.** The graphs depict IgG antibody levels against Rubella virus in sera of A, HC (n = 50) and COVID-19 cases (n = 50). B, HC divided into LTBi+ (n = 25) and LTBi- (n = 25) individuals. C, COVID-19 cases divided into those LTBi+ (n = 20) and LTBi– (n = 30). A dotted horizontal line indicates median value in the Violin plots. Cut off value of positivity for anti-Rubella IgG was 10 IU/ml. The Mann-Whitney U test was used to calculate the statistically significant differences (p ≤ 0.05) between groups.

comorbidities and LTBi. This may be due to the limited sample size of our cohort and the heterogeneity of COVID-19 cases studied, and therefore this requires further investigation.

The generation and maintenance of neutralizing antibodies against SARS-CoV-2 play an important role in resisting infection by the host. Antibodies against the Spike protein, especially the RBD of SARS-CoV-2, serve as targets for the development of vaccines and therapies [42]. Despite the reduction of MTB specific IFN-γ secreting cells in the COVID-19 group, there was a higher IgG response to RBD. The absence of any difference between IgG levels of LTBi positive and negative COVID-19 group participants suggests that humoral responses to infection were independent of the MTB-induced cellular activation observed. Our results are consistent with those reported by Silva et al who showed similarity in IgG levels to RBD between LTBi positive and negative individuals within COVID-19 groups [43]. Petron et al also reported LTBi positivity had no impact against SARS-CoV-2 virus with regard to the IgG response [44]. However, this is in contrast to another report which showed LTBi-COVID-19 co-infected individuals to have heightened levels of humoral, cytokine and acute phase responses compared to LTBi negative individuals [18].

Antibodies against Rubella were present in most participants in the study. Anti-Rubella IgG titers were similar in LTBi positive and negative individuals in both COVID-19 and HC groups. Thus, MTB exposure did not appear to impact the antibody levels against Rubella virus. Together, our results suggest that rather than priming for heterologous specific adaptive responses, trained innate responses triggered by the infection with MTB may account for the reduced susceptibility of LTBi individuals to SARS-CoV-2.

While comparing the mRNA levels of immune-related genes between HC and COVID-19 patients in unstimulated PBMCs, no significant difference was found for IFN-γ, IL-6, SOCS1, and IFN-1α in our study. However, we found higher levels of OAS-1, MAVS and SOCS3 in COVID-19 cases as compared to HC group. SOCS3 is cytokine regulator which negatively regulates cytokine signaling by binding to tyrosine kinase receptors, particularly through the JAK/STAT pathway [45]. Higher SOCS3 levels are also induced by pathogens to counter the host immune system against them [45]. MTB and SARS-CoV-2 both induce SOCS3 [46,47].

OAS1 and MAVS are Interferon Stimulated Genes (ISGs) which are up-regulated during active viral infections [48]. The higher levels of OAS1 and MAVS observed in COVID-19 indicates activation induced by SARS-CoV-2 whilst the absence

of differences between IFN-1α levels may be due to downregulation of T cell activation in this group. It is shown that SARS-CoV-2 can suppress the immune response either directly by depleting T cells [49] or by suppressing their effect by interfering with the synthesis of IFN-α and β [50].

In context of LTBi, here we did not find any differences in the expression of immune-related genes in HC or COVID-19 cases. However, while comparing LTBi positive individuals between HC and COVID-19, we found higher OAS-1 mRNA levels in the LTBi COVID-19 as compared to LTBi HC group.

### Limitations

We had aimed to recruit an equal number of HC and COVID-19 participants however, it was difficult to recruit patients due to their limited numbers at AKUH during the period of the study between 2022 and 2023 and their hesitation to be included in the study. Pakistan had relatively low COVID-19 morbidity and mortality during this period when omicron variants were circulating in the country [22,51]. It might be speculated that a decline in T cell responses or T cell exhaustion described in COVID-19 patients [52] can account for our observations. We found levels of both T-cell dependent specific or heterologous antibodies as well as the expression of inflammatory and type I IFN regulated genes similar in the LTBi positive and negative individuals. We acknowledge that it would be necessary to study levels of specific T cells recognizing SARS-CoV-2 in these groups remain to be determined to rule out the possibility of T cell exhaustion in this cohort. Unfortunately, we were unable to conduct flow cytometric analysis for T cell characterization in this study. Instead, to study the longer-term effect of COVID-19 we compared cellular responses in active and convalescent groups. Although these participants were recruited at different times after recovery, we were able to show a consistent result for MTB-stimulated IFN-γ SPUs as well as gene expression levels of OAS1, MAVS and SOCS3 between COVID-19 sub-groups in comparison with controls.

The understanding of 'TB infection' is currently evolving from being understood as a dichotomy to a continuum therefore, individuals identified with LTBi could possibly have subclinical TB. In low TB endemic countries, TB-immunoreactive individuals after 2–3 years of infection may clear MTB infection while preserving immunological memory of it. However, in high TB endemic countries infection can recur and lead to persistence of mycobacteria [11,53,54]. Therefore, any effect due to TB infection would be maintained despite the circulation of other pathogens such as observed in the pandemic.

### Conclusions

Understanding the impact of the pandemic in high TB burden setting is important. Our study shows milder COVID-19 in individuals with LTBi positive as compared with negative individuals. The modulation of host immunity after MTB exposure as well as differential upregulation of anti-viral genes in LTBi positive individuals likely reflects how SARS-CoV-2 may be restricted thereby limiting the severity of COVID-19. Further studies are required to investigate the combined impact of MTB and SARS-CoV-2 infections in the host.

### Supporting information

**S1 Fig. Increased ISG and SOCS3 mRNA expression in COVID-19 cases.**
(TIF)

**S2 Fig. Higher mRNA expression of OAS-1 among positive LTBi of COVID-19 cases compared to healthy controls.**
(TIF)

**S1 Table. Details of primers used for RT-PCR testing.**
(PDF)

**S2 Table. Comparison of characteristics of positive LTBi individuals in Healthy Control and COVID-19 groups.**
(PDF)

**S3 Table. mRNA expression in HC and COVID-19 study groups with and without LTBi.**
(PDF)

## Acknowledgements

Thanks to Joy Fleming and Lijun Bi of IBP, Chinese Academy of Sciences. We thank Ayesha Sadiqa for assisting with statistical analysis and Fizza Fatima for editorial review.

## Author contributions

**Conceptualization:** Uzair Abbas, Kiran Iqbal Masood, Martin Rottenberg, Rabia Hussain, Zahra Hasan.

**Data curation:** Uzair Abbas, Tulaib Iqbal, Bushra Jamil, Shama Qaiser, Maliha Yameen, Zahra Hasan.

**Formal analysis:** Uzair Abbas, Kiran Iqbal Masood, Tulaib Iqbal, Shama Qaiser, Maliha Yameen.

**Funding acquisition:** Zahra Hasan.

**Investigation:** Kiran Iqbal Masood.

**Methodology:** Kiran Iqbal Masood, Shama Qaiser, Martin Rottenberg, Rabia Hussain, Zahra Hasan.

**Project administration:** Zahra Hasan.

**Supervision:** Martin Rottenberg, Rabia Hussain, Zahra Hasan.

**Validation:** Kiran Iqbal Masood.

**Visualization:** Maliha Yameen.

**Writing – original draft:** Uzair Abbas, Kiran Iqbal Masood, Tulaib Iqbal, Bushra Jamil, Shama Qaiser, Maliha Yameen, Martin Rottenberg, Rabia Hussain, Zahra Hasan.

**Writing – review & editing:** Uzair Abbas, Kiran Iqbal Masood, Tulaib Iqbal, Bushra Jamil, Shama Qaiser, Maliha Yameen, Martin Rottenberg, Rabia Hussain, Zahra Hasan.

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
