## [Decision Letter · Decision Letter 0]

2 Jul 2025

Dear Dr. Hasan,

Thank you for submitting your manuscript to PLOS ONE. After careful consideration, we feel that it has merit but does not fully meet PLOS ONE’s publication criteria as it currently stands. Therefore, we invite you to submit a revised version of the manuscript that addresses the points raised during the review process.

**The reviewers have recommended publication, but also suggest significant revisions to your manuscript.  Therefore, I invite you to respond to the reviewers' comments and revise your manuscript.**

We look forward to receiving your revised manuscript.

Kind regards,

Fumihiro Yamaguchi

Academic Editor

PLOS ONE

**Journal Requirements:**

1. When submitting your revision, we need you to address these additional requirements. Please ensure that your manuscript meets PLOS ONE's style requirements, including those for file naming. The PLOS ONE style templates can be found at https://journals.plos.org/plosone/s/file?id=wjVg/PLOSOne_formatting_sample_main_body.pdf and https://journals.plos.org/plosone/s/file?id=ba62/PLOSOne_formatting_sample_title_authors_affiliations.pdf 2. During our evaluation of the documents provided, we noted that your ethics approval letter did not cover the entire date range for participant recruitment and data collection. Before we can proceed further with the submission, please provide the ethics approval extension document(s) for the study. If the document is in another language, please also provide an English translation. Please note that if this document is not included when your manuscript is resubmitted, it may be rejected. 3. Thank you for stating in your Funding Statement: TB Healthcare Biotechnology, China provided reagents and CDC, China provided support for the study. Uzair Abbas was supported by Higher Education Commission, Pakistan- Grand Challenges Fund Grant number -913, which also provided research support for this study.   Please provide an amended statement that declares *all* the funding or sources of support (whether external or internal to your organization) received during this study, as detailed online in our guide for authors at http://journals.plos.org/plosone/s/submit-now.  Please also include the statement “There was no additional external funding received for this study.” in your updated Funding Statement. Please include your amended Funding Statement within your cover letter. We will change the online submission form on your behalf. 4. Please include captions for your Supporting Information files at the end of your manuscript, and update any in-text citations to match accordingly. Please see our Supporting Information guidelines for more information: http://journals.plos.org/plosone/s/supporting-information.

Reviewers' comments:

Reviewer's Responses to Questions

**Comments to the Author**

1. Is the manuscript technically sound, and do the data support the conclusions?

Reviewer #1: Partly

Reviewer #2: Partly

2. Has the statistical analysis been performed appropriately and rigorously?

Reviewer #1: Yes

Reviewer #2: I Don't Know

3. Have the authors made all data underlying the findings in their manuscript fully available?

Reviewer #1: Yes

Reviewer #2: Yes

4. Is the manuscript presented in an intelligible fashion and written in standard English?

Reviewer #1: Yes

Reviewer #2: Yes

**Reviewer #1:**  Avoid Causal Language:

As this is a cross-sectional observational study, please avoid phrases like “LTBi confers protection” or “LTBi reduces severity.” Use terms such as “associated with” or “suggests a possible protective effect.”

Multivariate Analysis:

Include logistic regression to control for confounders such as age, sex, comorbidities, vaccination status, and time since infection. This is essential for substantiating your key conclusion.

Clarify LTBi Definition:

While the X.DOT-TB assay is explained, consider clarifying its comparability to standard QFT in the main Methods (not just Supplementary). It would also help to justify using only this assay for LTBi classification.

BCG Vaccination Interpretation:

BCG scar status was not associated with COVID-19 outcomes, but the Discussion still overemphasizes the “trained immunity” angle from BCG. Please reconcile this more clearly with your data.

Immunological Findings Interpretation:

The significance of raised OAS1 and SOCS3 in LTBi-positive COVID-19 cases is underexplained. Are these markers known to mediate antiviral responses in the TB context? Please elaborate or temper the discussion if speculative.

Discussion Length and Focus:

The Discussion repeats findings and prior literature excessively. Please condense and focus on the novel aspects of your findings.

**Reviewer #2: ** In this interesting paper the authors report a possible interaction between COVID-19 disease severity and LTBI status. They also observe limited or no effect of BCG status. This interaction may (partially) explain the epidemiology of SARS-Cov-2 in the Authors country.

I have one point regarding the interpretation of the results. The authors state that LTBi may protect against SARS-CoV-2 and reduce the severity of COVID-19 disease. I agree that the data presented appears to show that the severity of COVID-19 disease is reduced in persons with LTBi but I do not think the data presented convincingly shows protection against SARS-CoV-2 infection. Notably in figure 3 E compares IgG levels in a small number of HC who were either LTBi- (n=30) or LTBi+ (n=30), a significant proportion of the LTB+ve patients are positive for anti-RBD antibodies. The difference between the groups in figure 3E does not look significant to me and if I understand correctly both groups appear to have been approximately equally likely to have been infected with SARS-CoV-2 (and presumably had mild symptoms). If I am correct I would ask the authors to limit their claim of protection to protection from serious disease not protection from infection throughout the paper (introduction, discussion and conclusions). Notably also in the conclusions section of the abstract “We suggest that the immune reactivity to Mtb is associated with an increased protection against SARS-CoV-2 as indicated by both the lower frequency of COVID-19 and the reduced severity of disease in LTBi positive individuals.” delete “lower frequency of COVID-19” from this statement.

Minor point CODID should be COVID in the title “Higher numbers of M. tuberculosis-specific T cells in HC than in CODID 19 LTBi positive individuals.”

**Do you want your identity to be public for this peer review?** For information about this choice, including consent withdrawal, please see our Privacy Policy

Reviewer #1: No

Reviewer #2: **Yes: ** RM Anthony

---

## [Author Response · Author response to Decision Letter 1]

22 Jul 2025

Response to reviewers is attached in PDF file.

---

## [Decision Letter · Decision Letter 1]

28 Oct 2025

Dear Dr. Hasan,

Thank you for submitting your manuscript to PLOS ONE. After careful consideration, we feel that it has merit but does not fully meet PLOS ONE’s publication criteria as it currently stands. Therefore, we invite you to submit a revised version of the manuscript that addresses the points raised during the review process.

The reviewers have recommended publication, but also suggest significant revisions to your manuscript.  Therefore, I invite you to respond to the reviewers' comments and revise your manuscript.

We look forward to receiving your revised manuscript.

Kind regards,

Fumihiro Yamaguchi

Academic Editor

PLOS ONE

Journal Requirements:

Reviewers' comments:

Reviewer's Responses to Questions

**Comments to the Author**

Reviewer #2: All comments have been addressed

Reviewer #3: (No Response)

Reviewer #4: All comments have been addressed

2. Is the manuscript technically sound, and do the data support the conclusions?

Reviewer #2: Yes

Reviewer #3: Partly

Reviewer #4: Yes

3. Has the statistical analysis been performed appropriately and rigorously?

Reviewer #2: I Don't Know

Reviewer #3: Yes

Reviewer #4: Yes

4. Have the authors made all data underlying the findings in their manuscript fully available?

Reviewer #2: Yes

Reviewer #3: Yes

Reviewer #4: Yes

5. Is the manuscript presented in an intelligible fashion and written in standard English?

Reviewer #2: Yes

Reviewer #3: No

Reviewer #4: Yes

Reviewer #2: The authors have revised their paper and moderated the claims with respect to protection of infection and the potential protective effect of BCG. All the points raised have been addressed.

Reviewer #3: This study compares the Mycobacterium tuberculosis–specific cellular immune response in patients with COVID-19 versus healthy individuals. The authors conclude that this immune response is significantly more pronounced in healthy individuals compared with COVID-19 patients.

Major Comments

1. 1. The lack of interferon production by T cells indicates a decrease in cellular immunity, which may also be attributable to the higher prevalence of chronic comorbidities in the patient group compared with healthy controls. This should be clearly reflected in the discussion.

2. 2. The selection of patients and controls may be subject to biases, which are not sufficiently explained in the Materials and Methods section. For example, the timing of COVID-19 patient recruitment varies considerably after infection, which could substantially influence cellular immune responses.

3. 3. The fact that some patients were hospitalized while others were managed on an outpatient basis likely contributed to heterogeneous immune responses. This should be acknowledged and discussed as a limitation of the study.

4. 4. In the Methodology section, the process of recruiting both patients and controls requires a more straightforward explanation.

Minor Comments

5. 1. In the Abstract, the three introductory sentences are not entirely coherent with the primary study objective.

6. 2. The following sentence in the Abstract is unclear: "Therefore, MTB infection in active TB patients, or individuals with latent MTB infection, are likely to experience immune modulation that may alter the consequence of SARS-CoV-2 infection?"

7. 3. The authors should clarify what the messenger RNA sequences for OAS1, MAVS, SOCS1, and SOCS3 represent.

8. 4. The description of the mRNA sequences analyzed should be expanded, and the primers used for real-time PCR should be specified.

9. 5. For the method 'X. In the DOT-TB assay for screening LTBi, the antigens employed should be described.

10. 6. In the phrase 'As per manufacturer's instructions', the manufacturer of the kit must be explicitly named.

11. 7. In the Results section, subheadings currently present the outcomes rather than the study objectives.

12. 8. The entire manuscript requires a comprehensive revision of the English language, including correction of grammatical errors.

Reviewer #4: Overall, this is a well written revised manuscript with a few minor suggestions for revision;

INTRODUCTION:

- Rephrase 'with 677 million cases and 6.9 million deaths reported worldwide to 10

March 2023 ---' as 'with 677 million cases and 6.9 million deaths reported worldwide by 10

March 2023 ---'

- Define 'TST' at first use of term

- Rephrase 'Currently, studies on the association between SARSCoV-2 and MTB are both scare' as 'Currently, studies on the association between SARSCoV-2 and MTB are both scarce' ---

METHODS:

- You state that 'All 128 individuals in the COVID-19 group had been confirmed through a positive SARS-CoV-2 PCR

test'.

- Further, was genotyping performed to ascertain the SARS-CoV-2 strain that was associated with COVID-19 at that time?

- Further, for SARS-CoV-2 PCRs, did you use in-house or commercial kits?

- For RT-PCR on PBMCs, did you mean you did 'reverse-transcriptase real-time PCR' or just 'reverse-transcriptase PCR'? Please clarify.

- Are the primer sequences in-house, i.e., designed by you, or you used primer sequences published in literature?

DISCUSSION:

- In the limitations section, or somewhere in the discussion, comment on the fact that the phrase 'TB infection' is currently evolving from being understood as a dichotomy to a continuum, meaning the LTBi individuals could actually have been having TB disease? Just a thought.

**Do you want your identity to be public for this peer review?** For information about this choice, including consent withdrawal, please see our Privacy Policy

Reviewer #2: No

Reviewer #3: No

Reviewer #4: **Yes: ** David P Kateete

---

## [Author Response · Author response to Decision Letter 2]

14 Nov 2025

Dear Editor,

Thank you for giving us the opportunity to revise our manuscript. Please find below responses to comments made by the Reviewers.

Best wishes

Zahra Hasan

Response to Reviewers Comments

Reviewer #2: The authors have revised their paper and moderated the claims with respect to protection of infection and the potential protective effect of BCG. All the points raised have been addressed.

Ans. We thank reviewer 2 for their feedback and appreciation

Reviewer #3: This study compares the Mycobacterium tuberculosis–specific cellular immune response in patients with COVID-19 versus healthy individuals. The authors conclude that this immune response is significantly more pronounced in healthy individuals compared with COVID-19 patients.

Major Comments

1. 1. The lack of interferon production by T cells indicates a decrease in cellular immunity, which may also be attributable to the higher prevalence of chronic comorbidities in the patient group compared with healthy controls. This should be clearly reflected in the discussion.

Response:

Thank you for bringing up this point. We have referred to this point in the Discussion (l295-309).

2. 2. The selection of patients and controls may be subject to biases, which are not sufficiently explained in the Materials and Methods section. For example, the timing of COVID-19 patient recruitment varies considerably after infection, which could substantially influence cellular immune responses.

Response:

We have further explained the selection of patients and controls in the methods. The limitation of sample sizes of each cohort is discussed and as how the heterogeneity of the cohorts may contribute to the differences observed (l361-l376).

3. 3. The fact that some patients were hospitalized while others were managed on an outpatient basis likely contributed to heterogeneous immune responses. This should be acknowledged and discussed as a limitation of the study.

Response:

As mentioned above we have discussed this in the Limitations.

4. 4. In the Methodology section, the process of recruiting both patients and controls requires a more straightforward explanation.

Ans We have written up the process of recruitment of participants in more detail with inclusion and exclusion criteria for COVID-19 and Control groups.

Minor Comments

1. In the Abstract, the three introductory sentences are not entirely coherent with the primary study objective.

Response:

The abstract has been corrected.

2. The following sentence in the Abstract introduction is unclear: "Therefore, MTB infection in active TB patients, or individuals with latent MTB infection, are likely to experience immune modulation that may alter the consequence of SARS-CoV-2 infection?"

Response:

The abstract has been corrected.

3. The authors should clarify what the messenger RNA sequences for OAS1, MAVS, SOCS1, and SOCS3 represent.

Response: The sequences of primers used for RT-PCR to detect OAS-1, MAVS, SOCS1 and SOCS3 are given in S Table 1.

4. The description of the mRNA sequences analyzed should be expanded, and the primers used for real-time PCR should be specified.

Response:

The mRNA sequences were obtained from Eurofins (Fleuri, Luxembourg) and sequences were obtained from available literature and have been tested in our previous studies. The references and further details have been now mentioned in the methods.

5. For the method 'X. In the DOT-TB assay for screening LTBi, the antigens employed should be described.

Response:

The X DOT TB assay has ESAT 6 and CFP 10 antigens derived from MTB. This has been described in Method section.

6. In the phrase 'As per manufacturer's instructions', the manufacturer of the kit must be explicitly named.

Response:

TB Healthcare Biotechnology, China is the manufacturer of the kit. This has been mentioned now in the methods.

7. In the Results section, subheadings currently present the outcomes rather than the study objectives.

Response:

We have corrected subheadings in the Results section.

8. The entire manuscript requires a comprehensive revision of the English language, including correction of grammatical errors.

Response:

Thank you for your feedback. We have thoroughly revised the manuscript for English language and grammatical mistakes.

Reviewer #4: Overall, this is a well written revised manuscript with a few minor suggestions for revision;

INTRODUCTION:

1. Rephrase 'with 677 million cases and 6.9 million deaths reported worldwide to 10

March 2023 ---' as 'with 677 million cases and 6.9 million deaths reported worldwide by 10

March 2023 ---'

Response:

Thank you for highlighting the error. We have corrected the sentence as per the suggestion.

2. Define 'TST' at first use of term

Response:

We have defined Tuberculin Skin Test (TST) in the text.

3. Rephrase 'Currently, studies on the association between SARSCoV-2 and MTB are both scare' as 'Currently, studies on the association between SARSCoV-2 and MTB are both scarce'

Response:

We have replaced the word “scare with scarce”

METHODS:

4. You state that 'All 128 individuals in the COVID-19 group have been confirmed through a positive SARS-CoV-2 PCR test'. Further, was genotyping performed to ascertain the SARS-CoV-2 strain that was associated with COVID-19 at that time?

Response:

The SARS-CoV-2 strains from the participants in this cohort were not sequenced. However, at the time our group was involved in genomic surveillance, and we had reported that Omicron variants were predominant at the time of the study (1).

5. Further, for SARS-CoV-2 PCRs, did you use in-house or commercial kits?

Response:

No, the PCR was performed by the clinical laboratory of Aga Khan University Hospital and the cobas SARS-CoV-2 PCR assay was used, now detailed in Methods.

6. For RT-PCR on PBMCs, did you mean you did 'reverse-transcriptase real-time PCR' or just 'reverse-transcriptase PCR'? Please clarify.

Response:

Thank you we have clarified that we used reverse-transcriptase PCR using real time analysis for the study of mRNA expression of target genes. This has been clarified in Methods.

DISCUSSION:

8. In the limitations section, or somewhere in the discussion, comment on the fact that the phrase 'TB infection' is currently evolving from being understood as a dichotomy to a continuum, meaning the LTBi individuals could actually have been having TB disease? Just a thought.

Response:

Thank you for this point. We have added it to the Discussion and Conclusion.

7. Are the primer sequences in-house, i.e., designed by you, or did you use primer sequences published in literature?

Response:

The Methods section has additional information regarding the sources for the design of primers. Primers were procured from Eurofins (Fleuri, Luxembourg).

References

1. Bukhari AR, Ashraf J, Kanji A, Rahman YA, Trovao NS, Thielen PM, et al. Sequential viral introductions and spread of BA.1 across Pakistan provinces during the Omicron wave. BMC Genomics. 2023;24(1):432.

---

## [Decision Letter · Decision Letter 2]

3 Dec 2025

Individuals with latent tuberculosis in a high TB endemic country show mild COVID-19

PONE-D-25-25202R2

Dear Dr. Hasan,

We’re pleased to inform you that your manuscript has been judged scientifically suitable for publication and will be formally accepted for publication once it meets all outstanding technical requirements.

Kind regards,

Fumihiro Yamaguchi

Academic Editor

PLOS ONE

Additional Editor Comments (optional):

Reviewers' comments:

Reviewer's Responses to Questions

**Comments to the Author**

Reviewer #3: All comments have been addressed

2. Is the manuscript technically sound, and do the data support the conclusions?

Reviewer #3: Yes

3. Has the statistical analysis been performed appropriately and rigorously?

Reviewer #3: Yes

4. Have the authors made all data underlying the findings in their manuscript fully available?

Reviewer #3: Yes

5. Is the manuscript presented in an intelligible fashion and written in standard English?

Reviewer #3: Yes

Reviewer #3: The authors have fully addressed the reviewers’ comments and the article can be accepted for publication in its revised version

**Do you want your identity to be public for this peer review?** For information about this choice, including consent withdrawal, please see our Privacy Policy

Reviewer #3: No

---

## [Editor Report · Acceptance letter]

PONE-D-25-25202R2

PLOS One

Dear Dr. Hasan,

I'm pleased to inform you that your manuscript has been deemed suitable for publication in PLOS One. Congratulations! Your manuscript is now being handed over to our production team.

Kind regards,

on behalf of

Dr. Fumihiro Yamaguchi

Academic Editor

PLOS One